# Sketching Method for Large Scale Combinatorial Inference

**Will Wei Sun**
Department of Management Science
University of Miami
wsun@bus.miami.edu

**Junwei Lu**
Department of Biostatistics
Harvard University
junweilu@hsph.harvard.edu

**Han Liu**
Department of Computer Science
Northwestern University
hanliu@northwestern.edu

## Abstract

We present computationally efficient algorithms to test various combinatorial structures of large-scale graphical models. In order to test the hypotheses on their topological structures, we propose two adjacency matrix sketching frameworks: neighborhood sketching and subgraph sketching. The neighborhood sketching algorithm is proposed to test the connectivity of graphical models. This algorithm randomly subsamples vertices and conducts neighborhood regression and screening. The global sketching algorithm is proposed to test the topological properties requiring exponential computation complexity, especially testing the chromatic number and the maximum clique. This algorithm infers the corresponding property based on the sampled subgraph. Our algorithms are shown to substantially accelerate the computation of existing methods. We validate our theory and method through both synthetic simulations and a real application in neuroscience.

## 1 Introduction

Large-scale statistical inference has become a significant problem in era of big data especially in many scientific applications where both the scale of hypotheses to test and the sample size of data sets are significantly large [9]. One of the most important setting where the large-scale inference is applied to is the high dimensional graphical models, since they are useful to quantify the conditional dependency structure of a list of variables such as genes and brain neurons [20, 27].

There are two major challenges in the large-scale inference: statistical validity and computation efficiency. Most of the literature focus on the statistical validity of testing a large set of hypotheses. One of the criterion is to control the family-wise error of multiple hypothesis testing such that there is no type-II error [6, 26, 7]. In specific, [25] and [18] study the family-wise error control on the multiple hypotheses on the existence of edges in the graphical models. The other widely used statistical criterion in the large-scale hypothesis testing literature is the false discovery rate which aims to control the ratio between true discoveries and the number of hypotheses to test [3, 8, 2]. These large-scale inferential methods aim at inferring the edge-wise structure of graphs, while in many real applications it is more important to study the global *combinatorial* structures of graphs, for example, graph connectivity or maximal degree of the graph. [23] introduced statistical testings of the combinatorial structure of the underlying graph. To conduct the statistical testing, they proposed a two-step null-alternative witness technique by first identifying a minimal structure witnessing the alternative hypothesis, and then certifying the presence of this structure in the graph. Both of two

steps involve exhaustive structure searching and screening. In spite of the success in developing large-scale inference for graphical models with statistical guarantee, the second major challenge on how to develop computational efficient inferential method is less developed. Most of the methods above involve estimating the entire graph matrix which is computationally challenging if the number of vertices is extremely large. In particular, the combinatorial inference method proposed by [23] requires combinatorial searching for graph structures like loops, path and cliques. This is nearly computationally infeasible in testing graph properties like maximum clique and chromatic number.

In this paper we propose a novel combinatorial inferential framework with both the statistical validity and computational efficiency. A key ingredient of our method is that we consider two sketching strategies: neighborhood sketching and subgraph sketching. Both sketching methods aim to obtain a reduced network and thus we are able to conduct a refined inference on this reduced network. The idea of graph sketching was first initiated in the property testing literature [12, 13] in theoretical computer science. However, their approaches were applied to verify the combinatorial properties of deterministic graphs, which cannot be directly applied to the graphical models whose edge existence is uncertain. The refined inference step is then carefully performed on the network with relatively small size. The neighborhood sketching is proposed to infer the connectivity of graphs without estimating the entire graphical model. Due to computation and storage constraints, complete estimation of the graph will be infeasible in ultra-high dimensional models. However, we can still test the connectivity of the graph using our approach. The neighborhood sketching starts from randomly subsampling a few nodes in the network. We then run a breath-first neighborhood regression based on these sampled vertices. Comparing to the standard neighborhood regression method [21] which has the computation complexity $O(nd^2 \min\{n, d\})$ for a $d$-dimensional model with $n$ observations, our neighborhood sketching method has the complexity $\widetilde{O}(nd \min\{n, d\})$, where $\widetilde{O}(\cdot)$ means we ignore logarithmic terms. The subgraph sketching is designed to test the combinatorial structures which is NP hard to verify even for deterministic graphs. We focus on testing the $K$-colorability and maximal clique size in our paper. Comparing to the neighborhood sketching, the subgraph sketching method subsamples subgraphs and test structures which are critical to the topological structures. When testing the $K$-colorability, we subsample a set of subgraphs and test if they are $K$-colorable; and when testing the maximal clique size, we compute the clique density on the sampled subgraphs. The computation complexity can be greatly reduced through our approach. For example, the complexity of testing the $K$-colorability is reduced from $O(2.445^d)$ to $O(2.445^{K^2 \log K})$.

Besides the theoretical guarantees on the bounds of the type-I error and the power of the proposed algorithm, our paper also characterize the trade-off between the statistical validity and the computation complexity. We show that if the gap between null and alternative is larger, we can obtain a faster algorithm. Finally, our theory and method are validated through extensive experiments.

## 2 Combinatorial Inference via Sketching

In this section, we first formulate the inferential problems on Gaussian graphical model and then introduce the sketching-based algorithms for three large-scale combinatorial inference examples.

Let $\boldsymbol{X}_1, \ldots, \boldsymbol{X}_n \in \mathbb{R}^d$ be i.i.d. samples from $N_d(0, \boldsymbol{\Sigma})$. The corresponding precision matrix $\boldsymbol{\Theta} = \boldsymbol{\Sigma}^{-1}$ induces the conditional independence graph $G = (V, E)$, where the vertex set $V = \{1, \ldots, d\}$ and the edge set $E$ satisfies that an edge is present if and only if $\boldsymbol{\Sigma}_{jk} \neq 0$. The goal of combinatorial inference is to test where $G$ has certain global structures (e.g., connectivity) based on the random samples $\boldsymbol{X}_1, \ldots, \boldsymbol{X}_n$. In particular, let $\mathcal{G}$ be the set of graphs having a specific global structure, we aim to test the hypothesis $H_0$: G is of distance at least $\epsilon$ away from $\mathcal{G}$ versus $H_1$: $G \in \mathcal{G}$.

In the Gaussian graphical model, the testing of whether $G \in \mathcal{G}$ can be transferred to the testing on the precision matrix $\boldsymbol{\Theta} \in \mathcal{S}$, where $\mathcal{S} \subset \mathcal{M}(s)$ is the set of precision matrices such that for all $\boldsymbol{\Theta} \in \mathcal{S}$, we have $G(\boldsymbol{\Theta}) \in \mathcal{G}$. Here $\mathcal{M}(s)$ defines the parameter space of true precision matrices

$$\mathcal{M}(s) = \left\{ \boldsymbol{\Theta} \in \mathbb{R}^{d \times d} : \boldsymbol{\Theta} = \boldsymbol{\Theta}^\top, \lambda_{\min}(\boldsymbol{\Theta}) \geq 1/C, \|\boldsymbol{\Theta}\|_1 \leq L, \max_{j \in [d]} \|\boldsymbol{\Theta}_j\|_0 \leq s \right\} \quad (2.1)$$

for some constants $L > 0$ and $C \geq 1$. The above parameter space implies that the graphs considered are bounded degree graphs since there are at most $s$ nonzero entries for each column. In this paper, we focus on the large-scale graph scenario when the dimension $d$ is large and hence assume the true precision matrix to be sparse such that $\|\boldsymbol{\Theta}\|_1 \leq L$.

We formulate a procedure for testing $\boldsymbol{\Theta} \in \mathcal{S}$ using an estimated precision matrix $\widehat{\boldsymbol{\Theta}}$ obtained from samples $\boldsymbol{X}_1, \ldots, \boldsymbol{X}_n$. In particular, $\widehat{\boldsymbol{\Theta}}$ can be estimated via the node-wise regression [21], the CLIME [5], or the graphical lasso [11]. Although these estimators are consistent in parameter estimation, they are not directly applicable for statistical tests due to their non-ignorable bias terms [24]. Following [24], we introduce a de-biased estimator $\widehat{\boldsymbol{\Theta}}^d$ with its $(j, k)$-th component

$$\widehat{\boldsymbol{\Theta}}^d_{jk} = \widehat{\boldsymbol{\Theta}}_{jk} - \frac{\widehat{\boldsymbol{\Theta}}_j^\top (\widehat{\boldsymbol{\Sigma}}\widehat{\boldsymbol{\Theta}}_k - \boldsymbol{e}_k)}{\widehat{\boldsymbol{\Theta}}_j^\top \widehat{\boldsymbol{\Sigma}}_j}, \tag{2.2}$$

where $\widehat{\boldsymbol{\Sigma}}$ is the sample covariance matrix, $\widehat{\boldsymbol{\Theta}}_j$ is the $j$-th column of $\widehat{\boldsymbol{\Theta}}$, and $\boldsymbol{e}_k$ is a canonical unit vector with 1 at its $k$-th entry. Denote $c(\alpha, E)$ as the $(1 - \alpha)$ quantile estimator for the statistic $T_E := \max_{(j,k)\in E} \sqrt{n}(\widehat{\boldsymbol{\Theta}}^d_{jk} - \boldsymbol{\Theta}_{jk})$. The $\widehat{\boldsymbol{\Theta}}^d$ has strong control of the family-wise error rate.

**Lemma 2.1.** [[23]] Suppose that $\boldsymbol{\Theta} \in \mathcal{M}(s)$ and the precision matrix estimator $\widehat{\boldsymbol{\Theta}}$ satisfies $(A.1)$. If $(\log(dn))^7/n + s^2(\log(dn))^4/n \to 0$, for any edge set $E \subset V \times V$, we have, $\lim_{n\to\infty} \sup_{\boldsymbol{\Theta}} \mathbb{P}\left(\max_{(i,j)\in E} \sqrt{n}(\widehat{\boldsymbol{\Theta}}^d_{i,j} - \boldsymbol{\Theta}_{i,j}) > c(\alpha, E)\right) \leq \alpha$.

Here the conditions $(A.1)$ are satisfied by the aforementioned nose-wise regression, the CLIME estimator, and the graphical lasso estimator. Please see the supplementary for more details on conditions $(A.1)$ as well as the construction of the quantile estimator $c(\alpha, E)$.

The de-biased inference of graphical models mentioned above aims at inferring the edge-wise structure of graphs, while in many real applications it is more important to study the global combinatorial structures of graphs. Built upon this de-biased estimator $\widehat{\boldsymbol{\Theta}}^d$, in the following we will introduce our sketching-based inferential methods for testing three global combinatorial structures: connectivity, bipartite/K-colorability, and maximal clique, and study the theoretical properties for all of these tests.

Note that our inferential framework does not restrict to the Gaussian graphical model. It can also be utilized in general models, e.g., nonparanormal graphical model [17], transelliptical graphical model [16], and Ising model [22], as long as there exists a consistent precision matrix estimator.

## 2.1 Fast Connectivity Test

A graph $G = (V, E)$ is said to be connected if and only if there exists a path connecting each pair of its vertices. Denote the property *connectivity* as $\Pi$, and denote $\Pi_{d,s}$ as the class of connected graphs with $d$ vertices and bounded degree $s$. We are interested in the hypothesis testing:

$$H_0 : \text{G is disconnected with } \text{dist}(G, \Pi_{d,s}) \geq \epsilon, \quad H_1 : \text{G is connected.}$$

Here $\text{dist}(G, \Pi_{d,s})$ defines the distance between the graph $G$ and the set of connected graphs $\Pi_{d,s}$. Denote a function $f_G : V \times [d] \to V \cup \{0\}$ with $f_G(v, i) = u$ if $(u, v)$ is the i-th edge incident to $v$, and $f_G(v, i) = 0$ is there is no such edge. For two bounded degree graphs $G_1, G_2$ with node size $d$ and bounded degree $s$, following [13], we define the distance of two graphs as

$$\text{dist}(G_1, G_2) := \frac{|\{(v, i) : v \in [d], i \in [s], f_{G_1}(v, i) \neq f_{G_2}(v, i)\}|}{s \times d}. \tag{2.3}$$

Clearly, for any two graphs $G_1, G_2$, we have $0 \leq \text{dist}(G_1, G_2) \leq 1$. Intuitively, the numeric in $(2.3)$ quantifies 2 times of the number of edges needed to add in order to make $G_1$ and $G_2$ identical. Based on it, we further define the distance of a graph $G_1$ to a set of graph $\mathcal{C} = \{G : |V(G)| = d, \text{degree}(G) \leq s\}$ as $\text{dist}(G_1, \mathcal{C}) = \min_{G_2 \in \mathcal{C}} \text{dist}(G_1, G_2)$.

Now we are ready to introduce our sketching-based inferential method for the testing of connectivity in large-scale graphs. The algorithm of our fast connectivity test is shown in Algorithm 1.

**Remark 2.2.** In Algorithm 1, if the maximal degree parameter $s \geq 1$ is unknown, we can replace it via the estimator from the node-wise regression. It is important to mention that we do not need a consistent estimation of $s$. Instead, it is sufficient as long as the estimator is larger than the true sparsity, though a larger $s$ leads to less reduction in the computational cost. Moreover, the threshold $\tau_n$ plays an important role in the algorithm in order to obtain a theoretically guaranteed testing result. Following [24, 23], we consider $\tau_n = 0.5 \times \sqrt{\log d/n}$ in all our experiments.

---
**Algorithm 1** Fast Connectivity Test
---
1: **Input:** Samples $\boldsymbol{X}_1, \ldots, \boldsymbol{X}_n \in \mathbb{R}^d$, maximal degree $s$, distance $\epsilon$ in hypothesis, threshold $\tau_n$.
2: Compute the number of replicates $\ell = \min\{\lceil \log(8/(\epsilon s)) \rceil, d\}$.
3: **For** $i = 1$ to $\ell$ **Do**
4:     **Step 1:** Randomly select $m_i = \min\{\lceil \frac{32 \log(8/(\epsilon s))}{2^i \epsilon s} \rceil, d\}$ vertices in $1, \ldots, d$. Denote this set of vertices as $\mathcal{S}$.
5:     **Step 2: For** each vertex $j \in \mathcal{S}$ **Do**
6:         (1) Estimate $\widehat{\boldsymbol{\Theta}}_j$ via the node-wise regression and compute $\widehat{\boldsymbol{\Theta}}_j^d$ as in (2.2).
7:         (2) Select all $k \neq j$ such that $\mathcal{A}_j = \{k \neq j \mid |\widehat{\boldsymbol{\Theta}}_{jk}^d| \geq \tau_n\}$ and record the induced graph as $G_j$ whose vertices are $j \cup \mathcal{A}_j$ and edges are $\{e_{jk}, k \in \mathcal{A}_j\}$.
8:         (3) For each $k \in \mathcal{A}_j$, repeat (1) and (2) to update the induced graph $G_j$ by adding new vertices and edges. Continue this process until the leaf node is reached.
9:         (4) Performa a BFS on $G_j$ from (3) starting from $j$ until $2^i$ vertices have been selected or no new vertices can be reached. If it finds a small connected component, output ACCEPT.
10:     **End For**
11: **End For**
12: **Output:** If no small connected component is found in above search, output REJECT.
---

Below we analyze the computational complexity of the proposed fast connectivity test shown in Algorithm 1. According to [21], for each node-wise regression, the complexity is $O(nd \min\{n, d\})$. For each vertex $j$, the computational complexity in Step 2 is $O(nd \min\{n, d\}) + O(d) + 2^i[O(snd \min\{n, d\}) + O(d)] \sim 2^i O(snd \min\{n, d\})$. Therefore, the total computational complexity of our Algorithm 1 is $\sum_{i=1}^{\log(8/(\epsilon s))} m_i 2^i O(snd \min\{n, d\}) \sim O\left(nd \min\{n, d\} \log^2(1/(\epsilon s))/\epsilon\right) \sim \widetilde{O}\left(nd \min\{n, d\}/\epsilon\right)$, where $\widetilde{O}$ means up to a log term. On the other hand, directly estimating the whole precision matrix $\widehat{\Theta}$ needs to solve $d$ node-wise regression problems and have a total complexity $O(nd^2 \min\{n, d\})$. Clearly, when $\epsilon \succ 1/d$, our algorithm is faster than this direct approach. For instance, when $\epsilon = O(1)$, our rate is $\widetilde{O}\left(nd \min\{n, d\}\right)$ which is order $O(d)$ faster. This indicates the practical advantages of our algorithm in testing of large network where the vertex size $d$ is huge.

## 2.2 Fast Bipartiteness/K-colorability Test

In the testing of K-colorability, we would like to test if $K$ colors are sufficient to ensure that no two vertices sharing the same edge have the same color. The testing of bipartiteness is a special case of K-colorability test with $K = 2$. Our sketching-based inferential methods for testing these two properties share most of the steps with only slight difference in the parameter choice. However, the computational complexities of testing bipartiteness and K-colorability are largely different, where the former is polynomial and the latter with $K > 2$ is exponential. We are interested in the testing:

$$H_0 : \text{G is K-colorable}, \quad H_1 : \text{G is } \epsilon\text{-away from a K-colorable graph: dist}(G, \mathcal{G}_K) \geq \epsilon,$$

where $\mathcal{G}_K$ is the set of K-colorable graphs and the function $\text{dist}(\cdot, \cdot)$ is defined in (B.1). Here our null hypothesis is consistent with the null model in traditional statistical hypothesis tests by noting that an empty graph is clearly K-colorable.

Algorithm 2 summarizes our sketching-based inferential method for testing Bipartiteness and K-colorability. The computational complexity for each step of Algorithm 2 is as follows. **Step 1** takes $O(nd^2 \min\{n, d\})$ for node-wise regression and $O(d^3)$ for de-biased estimator; **Step 2** requires $O(m)$ opetations; **Step 3** requires $O(m^2)$, and **Step 4** requires $O(m \log m)$ for bipartiteness test [10] or $O(2.445^m)$ for the K-colorability test via dynamic programming [14].

**For bipartiteness test:** if we are given the de-biased estimator as in Step 1, then our algorithm has the advantage in the computation which reduces the complexity from $O(d^2)$ to $O(m^2)$. If we are not given the de-biased estimator, the complexity of the whole algorithm will be dominated by Step 1, and our algorithm has the same complexity as the direct approach.

**For K-colorability test:** The complexity of Algorithm 2 will be $O(2.445^m)$, which is smaller than the complexity $O(2.445^d)$ in the direct approach. Remind that $m = O(K^2 \log(K/\delta)/\epsilon^3)$. Therefore, up to a log term, our algorithm is faster than the direct approach when $\epsilon \succ d^{-1/3}$.

---

**Algorithm 2** Fast Bipartiteness/K-colorability Test

---

1: **Input:** Samples $\boldsymbol{X}_1, \ldots, \boldsymbol{X}_n \in \mathbb{R}^d$, distance $\epsilon$ in the hypothesis, confidence level $\alpha$.
2: **Step 1:** Estimate $\widehat{\boldsymbol{\Theta}}$ via the node-wise regression and compute $\widehat{\boldsymbol{\Theta}}^d$ as in (2.2).
3: **Step 2:** Set $\delta = \alpha/2$. Uniformly and randomly select $m := \min\{\lceil \log(1/(\epsilon\delta))/\epsilon^2 \rceil, d\}$ vertices for fast Bipartiteness test, or $m := \min\{\lceil K^2 \log(K/\delta)/\epsilon^3 \rceil, d\}$ vertices for fast K-colorability test. Denote this set of vertices as $\mathcal{S}$.
4: **Step 3:** For the edge set $E_\mathcal{S}$ of vertex set $\mathcal{S}$, compute $c(\alpha_1, E_\mathcal{S})$ as in $(A.2)$ with $\alpha_1 = \alpha/2$. Perform the following testing to obtain a reduced edge sets $\widetilde{E}_\mathcal{S}$,

$$\widetilde{E}_\mathcal{S} := \left\{ (j,k) \in E_\mathcal{S} | \sqrt{n} |\widehat{\boldsymbol{\Theta}}^d_{j,k}| > c(\alpha_1, E_\mathcal{S}) \right\}$$

and denote the induced graph as $G_\mathcal{S} := (\mathcal{S}, \widetilde{E}_\mathcal{S})$.
5: **Step 4:** Run DFS or BFS on the graph $G_\mathcal{S}$ to test if it is bipartite (or K-colorable).
6: **Output:** If **Step 4** is bipartite (or K-colorable), then output ACCEPT, otherwise REJECT.

---

## 2.3 Fast Max Clique Test

A clique is a subgraph such that any pair of its vertices are connected and the size of a clique is the number of vertices it contains. Find the maximal clique of a graph has many real applications, e.g., in social network we want to find a largest subset of people who all know each other; in protein-protein interaction network, we want to identify the group of proteins that connects to each other.

Denote $\omega(G)$ as the size of the largest clique of $G$, and denote the set $\mathcal{G}_\rho := \{G : \omega(G) \geq \rho|V(G)|\}$ as the set of graphs having cliques of density at least $\rho$. We are interested in the hypothesis testing:

$$H_0 : \text{G is } \epsilon\text{-away from } \mathcal{G}_\rho : \text{dist}(G, \mathcal{G}_\rho) \geq \epsilon, \quad H_1 : G \in \mathcal{G}_\rho.$$

Algorithm 3 summarizes our sketching-based inferential method for testing the size of max clique.

---

**Algorithm 3** Fast Max Clique Test (Density $\rho$)

---

1: **Input:** Samples $\boldsymbol{X}_1, \ldots, \boldsymbol{X}_n \in \mathbb{R}^d$, distance $\epsilon$ in the hypothesis, confidence level $\alpha$.
2: **Step 1:** Estimate $\widehat{\boldsymbol{\Theta}}$ via the node-wise regression and compute $\widehat{\boldsymbol{\Theta}}^d$ as in (2.2).
3: **Step 2:** Set $\delta = \alpha/2$. Uniformly and randomly select $m := \min\{\lceil 10q(\epsilon/2, \delta/5) \rceil, d\}$ vertices, where $q(\epsilon, \delta) := \log(1/\delta)/\epsilon$. Denote this set of vertices as $\mathcal{S}$.
4: **Step 3:** For the edge set $E_\mathcal{S}$ of vertex set $\mathcal{S}$, compute $c(\alpha_1, E_\mathcal{S})$ as in $(A.2)$ with $\alpha_1 = \alpha/2$. Perform the following testing to obtain a reduced edge sets $\widetilde{E}_\mathcal{S}$,

$$\widetilde{E}_\mathcal{S} := \left\{ (j,k) \in E_\mathcal{S} | \sqrt{n} |\widehat{\boldsymbol{\Theta}}^d_{j,k}| > c(\alpha_1, E_\mathcal{S}) \right\}$$

and denote the induced graph as $G_\mathcal{S} := (\mathcal{S}, \widetilde{E}_\mathcal{S})$.
5: **Step 4:** Apply Bron-Kerbosch algorithm [4] on $G_\mathcal{S}$ to compute the density of the maximal clique of $G_\mathcal{S}$.
6: **Output:** If the density from **Step 4** is at least $\rho - \epsilon/2$, output REJECT, otherwise ACCEPT.

---

Similar to the analysis in the fast K-colorability test, the complexity of Algorithm 3 will be dominated by the Bron-Kerbosch algorithm in Step 5, which is $O(3^{m/3})$ according to [29], which reduces the complexity $O(3^{d/3})$ in the direct approach. Therefore, up to a log term, our algorithm is faster than the direct approach by estimating the whole graph when $\epsilon \succ d^{-1}$.

## 3 Statistical Property

Our sketching-based inferential methods are not only efficient in the computational complexity, but also enjoy satisfactory statistical performance. In this section, we establish the upper bound of the type-I error and the lower bound of the power for each of the proposed algorithms.

Denote the minimal signal of the true precision matrix as $\beta_{\min} := \min_{(i,j) \in E^*} |\omega^*_{ij}|$. We start with the type-I error and power analysis of the proposed fast connectivity test introduced in Algorithm 1.

**Theorem 3.1.** The type-I error of the proposed fast connectivity test satisfies

$$\lim_{n\to\infty} \mathbb{P}(\text{Algorithm 1 outputs REJECT}|H_0) \le \alpha,$$

Moreover, if we assume $\beta_{\min} \ge 2\|\widehat{\Theta} - \Theta^*\|_{\max} = C\sqrt{\log d/n}$ for some constant $C$, we have

$$\lim_{n\to\infty} \{1 - \mathbb{P}(\text{Algorithm 1 outputs ACCEPT}|H_1)\} = 1.$$

Theorem 3.1 proves that the type-I error is well controlled by the confidence level $\alpha$ and the power is asymptotically one. This indicates that the sketching procedure used in our fast connectivity test does not cause any loss in the inferential accuracy.

Next result studies the type-I error and power analysis of the proposed fast bipartiteness/K-colorability test introduced in Algorithm 2.

**Theorem 3.2.** Assume the conditions in Lemma 2.1 are satisfied. The type-I error of the proposed fast bipartiteness/K-colorability test satisfies

$$\lim_{n\to\infty} \mathbb{P}(\text{Algorithm 2 outputs REJECT}|H_0) \le \alpha_1,$$

Moreover, if we assume $\beta_{\min} \ge C'\sqrt{\log d/n}$ for some large constant $C'$, then the power satisfies,

$$\lim_{n\to\infty} \{1 - \mathbb{P}(\text{Algorithm 2 outputs ACCEPT}|H_1)\} \ge (1-\alpha_1)(1-\delta).$$

where parameters $\alpha_1, \delta$ are defined in the Algorithm 2.

This lower bound of the power established in Theorem 3.2 indicates that the loss in power is due to two resources: the estimation procedure and the sketching procedure. In contrast to the connectivity test, the power loss in the fast bipartiteness/K-colorability test indicates a tradeoff in the computational cost and the inferential efficiency.

Finally, we analyze the inferential error of the fast max clique test introduced in Algorithm 3.

**Theorem 3.3.** Assume the conditions in Lemma 2.1 are satisfied. The type-I error of the proposed fast max clique test satisfies

$$\lim_{n\to\infty} \mathbb{P}(\text{Algorithm 3 outputs REJECT}|H_0) \le \alpha_1 + \delta,$$

Moreover, if we assume $\beta_{\min} \ge C'\sqrt{\log d/n}$ for some large constant $C'$, then the power satisfies,

$$\lim_{n\to\infty} \{1 - \mathbb{P}(\text{Algorithm 3 outputs ACCEPT}|H_1)\} \ge (1-\alpha_1)(1-\delta),$$

where parameters $\alpha_1, \delta$ are defined in the Algorithm 3.

Similar to the fast bipartiteness/K-colorability test, there is a loss in the power of the fast max clique test due to both the estimation and the sketching procedures. On the other hand, unlike the fast bipartiteness/K-colorability test, the sampling procedure also contributes to partial of the type-I error in the fast max clique test which implies a more stringent requirement on the sampling percentage.

## 4  Simulations

In this section, we compare our sketching-based inferential algorithm in Section 2.1 with some alternative solutions on the testing of connectivity. Due to space limit, additional simulation results for testing the K-colorability are included in the online supplement.

**Settings:** We investigate Algorithm 1 for various disconnected graphs under the null hypothesis. For each of 100 repetitions, we randomly remove $100 \times \gamma\%$ edges from a connected chain graph with $d$ vertices and denote its adjacency matrix as $\mathbf{A}_d(\gamma)$. We then generate i.i.d. samples $\mathbf{x}_i \sim N(0, (\Theta^*(\rho, \gamma))^{-1})$ for $i = 1, \ldots, n$, where $\Theta^*(\rho, \gamma) = \mathbb{1}_d + \rho \mathbf{A}_d(\gamma)$. Under the alternative hypothesis, we generate n i.i.d. samples $\mathbf{x}_i \sim N(0, (\Theta^*(\rho))^{-1})$ with $\Theta^*(\rho) = \mathbb{1}_d + \rho \mathbf{A}_d$, where $\mathbf{A}_d$ is the adjacency matrix of a connected chain graph with $d$ vertices. We consider settings with

$$n \in \{400, 800\}, d \in \{100, 200\}, \rho \in \{0.25, 0.3, 0.35\}, \gamma \in \{0.05, 0.275, 0.5\}.$$

**Methods:** We compare our algorithm, denoted as Ours, with 3 candidates: (1) Direct_clime: the method which estimates the whole precision matrix via CLIME first and then test the connectivity on the whole estimated graphs; (2) Direct_node: the method which estimates the whole precision matrix via nodewise regression first and then test the connectivity on the whole estimated graphs; (3) NLL: the two-step alternative witness test [23].

**Tuning Parameters:** Our algorithm depends on the degree $s$, the distance $\epsilon$, the truncation parameter $\tau$, and $\lambda$ in node-wise regression. Throughout all our experiments, we tune $\lambda$ in $\widehat{\boldsymbol{\theta}}_1$ via cross-validation and use the same $\lambda$ for the rest $\widehat{\boldsymbol{\theta}}_j$. We then estimate $s$ as $\widehat{s} = \|\widehat{\boldsymbol{\theta}}_1\|_0$ and estimate $\epsilon$ as $2/(\widehat{s} \times d)$. We use the theoretical rate for $\tau$ and set $\tau = 0.5\sqrt{\log(d)/n}$. For a fair comparison, the same tuning parameter $\lambda$ was used for Direct_node. In addition, we use $\lambda = 1.5\sqrt{\log(d)/n}$ for NLL as recommended in [23] and use the same tuning parameter for Direct_clime.

**Result 1 (Computational Costs):** Figure 1 illustrates the computational time of all methods. The left plot of Figure 1 shows computational time of our method over various disconnect ratio $\gamma$ in the scenario with $n = 400$, $d = 100$, and $\rho = 0.25$. It clearly illustrates that as the null graph gets more and more disconnected (larger $\gamma$), the computational time of our method gets shorter and shorter, which agrees with our theoretical complexity. Most importantly, as shown in the right plot of Figure 1, compared to the alternative methods, our method delivers clear advantages in computational costs. Due to the extremely large computational costs of NLL method [23], we only considered $d = 100$ and $d = 200$ in the simulations. As a reference, when $d = 500$, NLL method takes about 3.5 hours to run one replication while our method only takes less than 2 minutes.

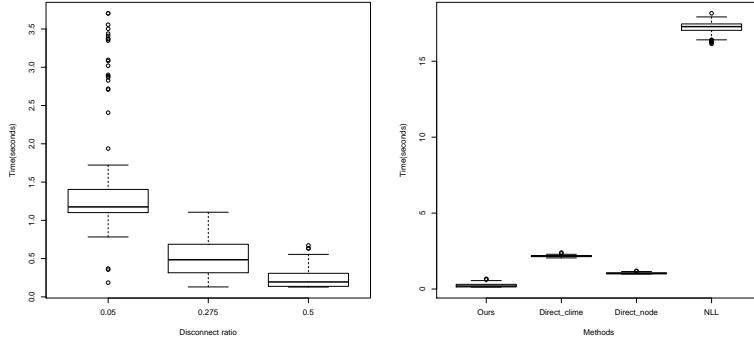

Figure 1: Left: computational time of our method over various disconnect ratio $\gamma$. Right: computational time of all 4 methods in the scenario with $n = 400$, $d = 100$, $\rho = 0.25$, and $\gamma = 0.5$.

**Result 2 (Type-I Error and Power):** Table 1 reports the type-I error as well as the power analysis of all methods in various scenarios with fixed $\gamma = 0.05$. When $\gamma = 0.275, 0.5$, the type-I errors of our method will all be zeros and hence the results are omitted. The type-I error of Direct_node is extremely large, which indicates that this method is too conservative in variable selection. Our method has a much smaller Type-I error than Direct_node and is slightly worse than Direct_clime and NLL. Moreover, the second part of Table 1 reports the power of all 4 methods in the connectivity testing. Our method has clear advantages over Direct_clime and NLL in the Power, especially when the signal level $\rho$ is small. Direct_node is too conservative in variable selection and hence it tends to include many noisy variables. Therefore, it is not surprised to see its power is large. In summary, Overall our method is able to deliver a satisfied inferential result with lowest computational costs. The other three competitive methods is either computational expensive (NLL), or has an extremely large Type-I error (Direct_node), or has a low power (Direct_clime or NLL).

## 5   Real Data Analysis

We apply our sketching-based inferential methods to an Neuroimaging study conducted by [28]. This study collects the fMRI scans of subjects who either listened to an intact story or a scrambled version of the story (the story was segmented into 608 short words and their order was scrambled randomly), and the goal of this study is to understand the difference of the brain images of these two groups. Both groups had 36 subjects and each subject had 300 fMRI measurements taken every

Table 1: The Type-I errors and powers of all 4 methods with fixed $\gamma = 0.05$. The best value for Type-I error is 0 and the best value for the power is 1.

| n | d | Methods | Type-I error | | | Power | | |
|---|---|---|---|---|---|---|---|---|
| | | | $\rho = 0.25$ | $\rho = 0.3$ | $\rho = 0.35$ | $\rho = 0.25$ | $\rho = 0.3$ | $\rho = 0.35$ |
| 400 | 100 | Ours | 0.19 | 0.12 | 0.11 | 0.67 | 0.98 | 1 |
| | | Direct_clime | 0 | 0 | 0 | 0 | 0 | 0.76 |
| | | Direct_node | 0.98 | 1 | 0.97 | 1 | 1 | 1 |
| | | NLL | 0 | 0 | 0 | 0 | 0 | 0.23 |
| 400 | 200 | Ours | 0.06 | 0.06 | 0.08 | 0.07 | 0.93 | 1 |
| | | Direct_clime | 0 | 0 | 0 | 0 | 0 | 0.16 |
| | | Direct_node | 0.96 | 0.98 | 0.99 | 0.99 | 1 | 1 |
| | | NLL | 0 | 0 | 0 | 0 | 0 | 0.01 |
| 800 | 100 | Ours | 0.14 | 0.09 | 0.13 | 1 | 1 | 1 |
| | | Direct_clime | 0 | 0 | 0 | 0.55 | 1 | 1 |
| | | Direct_node | 0.99 | 0.97 | 0.98 | 1 | 1 | 1 |
| | | NLL | 0 | 0 | 0 | 0.02 | 0.76 | 1 |
| 800 | 200 | Ours | 0.07 | 0.08 | 0.03 | 0.99 | 1 | 1 |
| | | Direct_clime | 0 | 0 | 0 | 0.04 | 1 | 1 |
| | | Direct_node | 0.96 | 0.97 | 0.98 | 1 | 1 | 1 |
| | | NLL | 0 | 0 | 0 | 0 | 0.45 | 0.98 |

1.4 seconds. For both the intact story group and the scrambled word group, we average the datasets across subjects, and finally obtain two data matrices of dimension $n = 300$ and $d = 823$. [19] employed the combinatorial inference to this data set to study the difference in the connectivity levels and the maximum degrees of the brain networks in the intact story group and the scrambled word group. In this paper, we focus on two tasks of the maximum clique test. The first one is to compare the computational cost and the statistical testing accuracy of our fast max clique test with those of the max clique test applied directly on the original data (Direct). The second one is to evaluate the difference of maximal cliques generated from the intact story group and the word scrambled group.

In our algorithm, we set the parameter $\epsilon = 0.24$ and the confidence level $\alpha = 0.05$. As a result, our test sampled 625 nodes out of total 823 regions. Such sampling leads to a huge reduction in computation. In particular, the Direct method takes about 2649 seconds to compute the maximal clique, while Ours only takes about 299 seconds. Such computational reduction implies the importance of sampling in testing large graphs. Importantly, we do not sacrifice testing accuracy for such computational cost. Figure 2 reports the max cliques identified by Direct method and Ours method for the intact story group and the word scrambled group. Clearly, for each group, our fast algorithm is able to find almost all regions that contribute to the max clique in the original graph. In the second task, as shown in Figure 2, max cliques found in our fast max clique test locate primarily in the precuneus region, which is well known for understanding high-level concepts in stories [1]. Moreover, our fast test evaluates that the max clique density of the intact story group is larger than that of the word scrambled group. This is consistent with the findings in brain literatures that the brain connectivity in the precuneus region tends to be more active in the intact story group [28, 19].

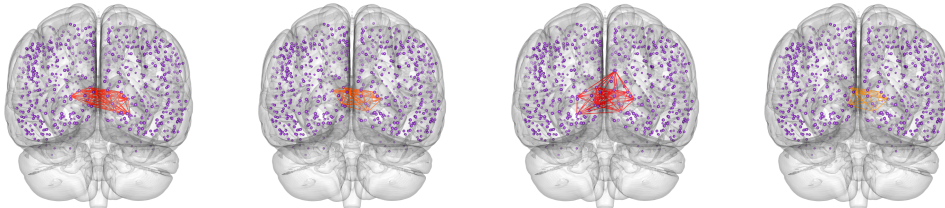

Figure 2: The max cliques identified by the Direct method for the intact story group and the word scrambled group are shown in the first and second plot, respectively. The max cliques identified by our method for the intact story group and the word scrambled group are shown in the third and forth plot, respectively. In each plot, the darker the color, the larger the clique density is.

**Acknowledgments**

Han Liu's research is supported by the NSF BIGDATA 1840866, NSF RI 1408910, NSF CAREER 1841569, NSF TRIPODS 1740735, along with an Alfred P Sloan Fellowship.

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
