[Supplementary Material]

# Online Supplement for
# Sketching Method for Large Scale Combinatorial Inference

In this appendix, we review the quantile estimation for graphical models, introduce a few important lemmas in property testing literatures, then provide detailed proofs of our main theorems in our fast tests of connectivity, bipartiteness, K-colorability, and max-cliqueness, and finally discuss additional simulation studies.

## A    Quantile Estimation for Graphical Models

It is worthy mentioning that our large-scale combinatorial inference does not reply on that $\widehat{\Theta}$ is the node-wise regression estimator. Let $\widehat{\Sigma}$ be the sample covariance matrix. In general, we only require the estimator satisfies the following generic conditions:

$$\|\widehat{\Theta} - \Theta\|_{\max} \le C\sqrt{\log d/n}, \|\widehat{\Theta} - \Theta\|_1 \le Cs\sqrt{\log d/n}, \|\widehat{\Sigma}\widehat{\Theta} - I_d\|_{\max} \le C\sqrt{\log d/n}. \quad \text{(A.1)}$$

These generic conditions are satisfied by a wide list of estimators, including the aforementioned node-wise regression estimator [21], the CLIME estimator [5], the graphical lasso estimator [11], and the Gaussian copula graphical model [15]. Based on this generic estimator $\widehat{\Theta}$, we follow [24] to construct the $(1 - \alpha)$ quantile estimator $c(\alpha, E)$ of the statistic $T_E := \max_{(j,k)\in E} \sqrt{n}(\widehat{\Theta}_{jk}^d - \Theta_{jk})$. We use the Gaussian multiplier bootstrap procedure to construct

$$T^B := \sup_{(j,k)\in E} \frac{1}{\sqrt{n}} \sum_{i=1}^{n} \widehat{\Theta}_j^\top (X_i X_i^\top \widehat{\Theta}_k - e_k)\psi_i$$

where $\psi_i, i = 1, \ldots, n$ are i.i.d. standard normal random variables. Finally, we estimate the conditional quantile of $T_E$ by

$$c(\alpha, E) = \inf\left\{ t \in \mathbb{R} | \mathbb{P}\left(|T^B| > t | \{X_i\}_{i=1}^n\right) \le \alpha \right\} \quad \text{(A.2)}$$

## B    Auxiliary lemmas

The following lemma quantifies the number of connected components, i.e., the maximal connected subgraphs, of a graph when its whole graph is not connected. This lemma will be used in the type-I error analysis of our fast connectivity test.

**Lemma B.1.** [Corollary 3.3, [13]] If a graph $G$ is $\epsilon$-far from the set of $d$-vertex connected graphs of bounded degree $s \ge 2$, then $G$ has at least $\frac{\epsilon ds}{8}$ connected components each containing less than $\frac{8}{\epsilon s}$ vertices.

Before stating lemmas for property tests of K-colorability and max-clique, we introduce a new distance function for them [12], which is parallel to (2.3) for the connectivity test.

$$\text{dist}(G_1, G_2) := \frac{|(E(G_1)\backslash E(G_2)) \cup (E(G_2)\backslash E(G_1))|}{d \times d}. \quad \text{(B.1)}$$

The distance definitions of (2.3) and (B.1) are suitable for different property tests. For instance, (2.3) was used in [13] for the connectivity test, and on the other hand, (B.1) was employed in [12] for the K-colorable test and max clique test. We follow this convention in this paper. The next lemma shows that the K-colorability testing algorithm proposed in [12] is a property testing algorithm with well-controlled errors. Their K-colorability testing algorithm first uniformly chooses a set of $O(K^2 \log(K/\delta)/\epsilon^3)$ vertices, then tests if the reduced graph is K-colorable or not.

**Lemma B.2.** [Theorem 6.2.2, [12]] If a graph $G$ is K-colorable, then it is accepted by above K-colorability testing algorithm with probability 1. Moreover, every graph $G$ which is $\epsilon$-far from the class of K-colorable graphs is rejected by the K-colorability testing algorithm with probability at least $1 - \delta$.

The next lemma shows the property of sampling vertices in the max-clique test.

**Lemma B.3.** [Corollary 7.2, [12]] If a graph $G$ has cliques of density at least $\rho$, then with probability at least $1 - \delta$, the reduced graph from an uniformly chosen set of $\text{poly}(\epsilon^{-1}\log(1/\delta))$ vertices has a clique of density at least $\rho - \epsilon/2$. Moreover, if a graph $G$ is $\epsilon$-far from the class of graphs having cliques of density at least $\rho$, then with probability at most $\delta$, the reduced graph from an uniformly chosen set of $\text{poly}(\epsilon^{-1}\log(1/\delta))$ vertices has a clique of density at least $\rho - \epsilon/2$.

## C  Proof of Theorem 3.1

We compute the Type I and Type II errors of the proposed algorithm. We show that as $n \to \infty$,

$$\text{Type-I Error: } \mathbb{P}(\text{Algorithm outputs REJECT}|H_0) \leq \alpha,$$
$$\text{Type-II Error: } \mathbb{P}(\text{Algorithm outputs ACCEPT}|H_1) = 0.$$

Based on Lemma B.1, if $G$ is $\epsilon$-far from the class of connected graphs, then the probability that a uniformly selected vertex belongs to a connected component which contains less than $8/(\epsilon s)$ vertices, is at least $\frac{\epsilon ds/8}{d} = \frac{\epsilon s}{8}$. Therefore, if we uniformly select $m$ vertices, the probability that no selected vertex belongs to a component of size less than $\frac{\epsilon s}{8}$ is bounded above by

$$\left(1 - \frac{\epsilon s}{8}\right)^m < \exp(-\frac{\epsilon s m}{8}).$$

Therefore, given a required value $\beta$ for this upper bound, we can compute the needed size for random sampling, that is,

$$m \geq \frac{8\log(\beta)}{\epsilon s}.$$

**Type-I Error**: In order to ensure that Type-I error is bounded by a pre-specified constant $\alpha$, we need to control

$$\mathbb{P}(\text{Algorithm outputs ACCEPT}|H_0) \geq 1 - \alpha.$$

According to Lemma B.1, we only need to ensure

$$1 - \left[1 - \frac{2^i \epsilon s}{16\log(8/(\epsilon s))}\right]^{m_i} \geq 1 - \alpha.$$

According to the inequality $(1 - a)^x < \exp(-ax)$ for any $0 < a < 1$ and $x \in \mathbb{R}$, it is sufficiently to require

$$m_i \geq \frac{16\ln(1/\alpha)\log(8/(\epsilon s))}{2^i \epsilon s},$$

Note that the $m_i$ used in the algorithm corresponds to the case with $\alpha = 1/3$. In practice, it can be adjusted accordingly for a pre-specified $\alpha$.

**Type-II Error**: The Type-II error of the proposed algorithm is always 0. This holds because the variable selection consistency $\text{supp}(\widehat{\Theta}_j) = \text{supp}(\Theta_j^*)$ for each $j = 1, \ldots, d$ holds almost surely under the minimal signal assumption. This completes the proof of Theorem 3.1.

## D  Proof of Theorem 3.2

We compute the Type I error and Power of the proposed algorithm separately. We show that, as $n \to \infty$,

$$\text{Type-I Error:} \quad \mathbb{P}(\text{Algorithm outputs REJECT}|H_0) \leq \alpha,$$
$$\text{Power:} \quad \mathbb{P}(\text{Algorithm outputs REJECT}|H_1) \geq (1 - \alpha_1)(1 - \delta).$$

**Type-I Error**: Denote the set of graphs under $H_0$ as $\mathcal{H}_0 := \{G : \text{G is K-colorable}\}$. In order to ensure that Type-I error is bounded by $\alpha_1$, we need to control

$$\sup_{G^* \in \mathcal{H}_0} \mathbb{P}(\text{Algorithm outputs REJECT}) \leq \alpha_1.$$

By the construction of the algorithm, it is sufficient to prove that

$$\sup_{G^* \in \mathcal{H}_0} \mathbb{P}(G_{\mathcal{S}} \text{ is not bipartite/K-colorable}) \leq \alpha_1.$$

where $G_{\mathcal{S}} = (\mathcal{S}, \widetilde{E}_{\mathcal{S}})$ as defined in the algorithm.

Denote the true edge set induced by the vertex $\mathcal{S}$ as $E_{\mathcal{S}}^*$ and the corresponding true graph as $G_{\mathcal{S}}^* := (\mathcal{S}, E_{\mathcal{S}}^*)$. We next show that

$$\lim_{n \to \infty} \mathbb{P}\left(\widetilde{E}_{\mathcal{S}} \subset E_{\mathcal{S}}^*\right) \geq 1 - \alpha_1. \tag{D.1}$$

For the edge set $E_{\mathcal{S}}$ defined based on sampled vertices in Step 4 of the algorithm, if we let $E = E_{\mathcal{S}}$ in Lemma 2.1, we have, uniformly over $\boldsymbol{\Theta} \in \mathcal{M}(s)$,

$$\lim_{n \to \infty} \mathbb{P}\left(\max_{(i,j) \in E_{\mathcal{S}}} \sqrt{n}(\widehat{\boldsymbol{\Theta}}_{i,j}^d - \boldsymbol{\Theta}_{i,j}) > c(\alpha_1, E_{\mathcal{S}})\right) \leq \alpha_1.$$

Denote the true null set as $E^n$. Therefore, the reduced edge set $\widetilde{E}_{\mathcal{S}} := \left\{(j,k) \in E_{\mathcal{S}} | \sqrt{n}|\widehat{\boldsymbol{\Theta}}_{j,k}^d| > c(\alpha, E_{\mathcal{S}})\right\}$ defined in Step 4 of the algorithm satisfies, if $(i,j) \in E_{\mathcal{S}} \cup E^n$, then $\boldsymbol{\Theta}_{i,j} = 0$, and $\max_{(i,j) \in E_{\mathcal{S}}} \sqrt{n}|\widehat{\boldsymbol{\Theta}}_{i,j}^d| \leq c(\alpha, E_{\mathcal{S}})$, and hence $(i,j) \notin \widetilde{E}_{\mathcal{S}}$. That is, as $n \to \infty$, for any edge $(i,j) \in E_{\mathcal{S}}$, if $(i,j) \notin E_{\mathcal{S}}^*$, then $(i,j) \notin \widetilde{E}_{\mathcal{S}}$ with probability at least $1 - \alpha_1$, uniformly over $\boldsymbol{\Theta} \in \mathcal{M}(s)$. This validates (D.1).

In addition, by the monotonic property of the bipartiteness and K-colorability, we have if a graph $G^* = (V, E^*)$ is bipartite/K-colorable, then its subgraph $G_{\mathcal{S}}^* = (\mathcal{S}, E_{\mathcal{S}}^*)$ must be a bipartite/K-colorable graph. Therefore,

$$\sup_{G^* \in \mathcal{H}_0} \mathbb{P}(G_{\mathcal{S}}^* \text{ is not bipartite/K-colorable}) = 0. \tag{D.2}$$

Combining (D.1) and (D.2) will lead to the desirable results. In particular, we have, as $n \to \infty$,

$$\sup_{G^* \in \mathcal{H}_0} \mathbb{P}(G_{\mathcal{S}} \text{ is not bipartite/K-colorable})$$

$$= \sup_{G^* \in \mathcal{H}_0} \left\{ \mathbb{P}(G_{\mathcal{S}} \text{ is not bipartite/K-colorable} | \widetilde{E}_{\mathcal{S}} \subset E_{\mathcal{S}}^*) * \mathbb{P}(\widetilde{E}_{\mathcal{S}} \subset E_{\mathcal{S}}^*) \right.$$

$$\left. + \mathbb{P}(G_{\mathcal{S}} \text{ is not bipartite/K-colorable} | \widetilde{E}_{\mathcal{S}} \not\subset E_{\mathcal{S}}^*) * \mathbb{P}(\widetilde{E}_{\mathcal{S}} \not\subset E_{\mathcal{S}}^*) \right\}$$

$$\leq \sup_{G^* \in \mathcal{H}_0} \mathbb{P}(G_{\mathcal{S}} \text{ is not bipartite/K-colorable} | \widetilde{E}_{\mathcal{S}} \subset E_{\mathcal{S}}^*) + \sup_{G^* \in \mathcal{H}_0} \mathbb{P}(\widetilde{E}_{\mathcal{S}} \not\subset E_{\mathcal{S}}^*)$$

$$\leq 0 + \alpha_1 \leq \alpha_1,$$

where the second inequality is due to the two arguments shown in (D.1) and (D.2). This upper bound $\alpha_1$ indicates that the Type-I error is only due to the estimation error from Lemma 2.1 and there will be no loss in the Type-I error due to the sampling procedure.

**Power**: In order to derive the lower bound of Power, it is equivalent to show

$$\mathbb{P}(G_{\mathcal{S}} \text{ is not bipartite/K-colorable} | H_1) \geq (1 - \delta)(1 - \alpha_1).$$

Denote the true edge set induced by the vertex $\mathcal{S}$ as $E_{\mathcal{S}}^*$ and the corresponding true graph as $G_{\mathcal{S}}^* := (\mathcal{S}, E_{\mathcal{S}}^*)$. According to Lemma 2.1 and the minimal signal condition on $\beta_{\min}$, we have

$$\lim_{n \to \infty} \mathbb{P}\left(\widetilde{E}_{\mathcal{S}} = E_{\mathcal{S}}^*\right) \geq 1 - \alpha_1. \tag{D.3}$$

Moreover, Lemma B.2 implies that if algorithm accepts $G_{\mathcal{S}}^*$ with probability at least $\delta$, then the true original graph $G^*$ must be $\epsilon$-far from to the set of K-colorable graph $\mathcal{G}_K$, that is,

$$\mathbb{P}(\text{Algorithm rejects } G_{\mathcal{S}}^* | H_1) \geq 1 - \delta. \tag{D.4}$$

Therefore, as $n \to \infty$, we have

$$\mathbb{P}(G_{\mathcal{S}} \text{ is not bipartite/K-colorable} | H_1) =$$

$$= \mathbb{P}(G_{\mathcal{S}} \text{ is not bipartite/K-colorable} | H_1, G_{\mathcal{S}} = G_{\mathcal{S}}^*) \mathbb{P}(G_{\mathcal{S}} = G_{\mathcal{S}}^*)$$

$$+ \mathbb{P}(G_{\mathcal{S}} \text{ is not bipartite/K-colorable} | H_1, G_{\mathcal{S}} \neq G_{\mathcal{S}}^*) \mathbb{P}(G_{\mathcal{S}} \neq G_{\mathcal{S}}^*)$$

$$\geq \mathbb{P}(G_{\mathcal{S}} \text{ is not bipartite/K-colorable} | H_1, G_{\mathcal{S}} = G_{\mathcal{S}}^*) \mathbb{P}(G_{\mathcal{S}} = G_{\mathcal{S}}^*)$$

$$\geq \mathbb{P}(\text{Algorithm rejects } G_{\mathcal{S}}^* | H_1) \mathbb{P}(G_{\mathcal{S}} = G_{\mathcal{S}}^*)$$

$$\geq (1 - \delta)(1 - \alpha_1),$$

where the second inequality is due to the arguments shown in $(D.3)$ and $(D.4)$. This lower bound of Power indicates the power loss is due to two resources: the estimation procedure and the sampling procedure. The partial power loss due to sampling procedure indicates a tradeoff in the computational cost and the inferential efficiency. This finishes the proof of Theorem 3.2.

## E  Proof of Theorem 3.3

We compute the Type I error and Power of the proposed algorithm separately. We show that, as $n \to \infty$,

$$\text{Type-I Error:} \quad \mathbb{P}(\text{Algorithm outputs REJECT}|H_0) \leq \alpha,$$
$$\text{Power:} \quad \mathbb{P}(\text{Algorithm outputs REJECT}|H_1) \geq (1 - \alpha_1)(1 - \delta).$$

**Type-I Error**: Denote the set of graphs under $H_0$ as $\mathcal{H}_0 := \{G : \text{dist}(G, \mathcal{G}_\rho) \geq \epsilon\}$. In order to ensure that Type-I error is upper bounded by a pre-specified constant $\alpha_1$, by the construction of the algorithm, it is equivalent to show that

$$\sup_{G^* \in \mathcal{H}_0} \mathbb{P}(G_\mathcal{S} \text{ has density} \geq \rho - \epsilon/2) \leq \alpha_1.$$

For the edge set $E_\mathcal{S}$ defined based on sampled vertices in Step 4 of the algorithm, if we let $E = E_\mathcal{S}$ in Lemma 2.1, we have, uniformly over $\boldsymbol{\Theta} \in \mathcal{M}(s)$,

$$\lim_{n \to \infty} \mathbb{P}\left( \max_{(i,j) \in E_\mathcal{S}} \sqrt{n}(\widehat{\boldsymbol{\Theta}}_{i,j}^d - \boldsymbol{\Theta}_{i,j}) > c(\alpha_1, E_\mathcal{S}) \right) \leq \alpha_1.$$

By a similar argument as in $(D.1)$ for the K-colorable case, we have, as $n \to \infty$, for any edge $(i,j) \in E_\mathcal{S}$, if $(i,j) \notin E_\mathcal{S}^*$, then $(i,j) \notin \widetilde{E}_\mathcal{S}$ with probability at least $1 - \alpha_1$, uniformly over $\boldsymbol{\Theta} \in \mathcal{M}(s)$. Therefore, we also have,

$$\lim_{n \to \infty} \mathbb{P}\left( \widetilde{E}_\mathcal{S} \subset E_\mathcal{S}^* \right) \geq 1 - \alpha_1. \tag{E.1}$$

Moreover, by the monotonic property of the max clique, for any $\rho > 0$, if a graph $G_\mathcal{S} = (\mathcal{S}, \widetilde{E}_\mathcal{S})$ is has a max clique density $\rho$, then we must have its superset $G_\mathcal{S}^* = (\mathcal{S}, E_\mathcal{S}^*)$ also has a max clique density at lest $\rho$. Therefore,

$$\mathbb{P}(G_\mathcal{S}^* \text{ has density} \geq \rho - \epsilon/2) \geq \mathbb{P}(G_\mathcal{S} \text{ has density} \geq \rho - \epsilon/2). \tag{E.2}$$

Moreover, Lemma B.3 implies that, if $\text{dist}(G, \mathcal{G}_\rho) \geq \epsilon$, then

$$\mathbb{P}(G_\mathcal{S}^* \text{ has density} \geq \rho - \epsilon/2) < \delta. \tag{E.3}$$

Therefore, we have, as $n \to \infty$,

$$\sup_{G^* \in \mathcal{H}_0} \mathbb{P}(G_\mathcal{S} \text{ has density} \geq \rho - \epsilon/2)$$
$$\leq \sup_{G^* \in \mathcal{H}_0} \mathbb{P}(G_\mathcal{S} \text{ has density} \geq \rho - \epsilon/2 | \widetilde{E}_\mathcal{S} \subset E_\mathcal{S}^*) \mathbb{P}(\widetilde{E}_\mathcal{S} \subset E_\mathcal{S}^*) + \sup_{G^* \in \mathcal{H}_0} \mathbb{P}(\widetilde{E}_\mathcal{S} \not\subset E_\mathcal{S}^*)$$
$$\leq \sup_{G^* \in \mathcal{H}_0} \mathbb{P}(G_\mathcal{S}^* \text{ has density} \geq \rho - \epsilon/2) + \sup_{G^* \in \mathcal{H}_0} \mathbb{P}(\widetilde{E}_\mathcal{S} \not\subset E_\mathcal{S}^*)$$
$$\leq \delta + \alpha_1,$$

where the second inequality is due to $(E.2)$ and the last inequality is due to the arguments in $(E.3)$ and $(E.1)$. This indicates that the Type-I error is due to the addition of two resources: the estimation procedure $(\alpha_1)$ and the sampling procedure $(\delta)$.

**Power**: In order to find the lower bound of Power, we need to control

$$\mathbb{P}(\text{Algorithm outputs REJECT}|H_1).$$

According to Lemma 2.1 and the minimal signal condition on $\beta_{\min}$, we have

$$\lim_{n \to \infty} \mathbb{P}\left( \widetilde{E}_\mathcal{S} = E_\mathcal{S}^* \right) \geq 1 - \alpha_1. \tag{E.4}$$

Moreover, Lemma B.3 implies that, if $G^* \in \mathcal{G}_\rho$, then its population-version subgraph $G_{\mathcal{S}}^*$ corresponding to the sample-version $G_{\mathcal{S}}$ in the algorithm, satisfies,

$$\mathbb{P}(G_{\mathcal{S}}^* \text{ has density} \geq \rho - \epsilon/2) < \alpha_1. \tag{E.5}$$

Note that, as $n \to \infty$, we have

$$
\begin{aligned}
\mathbb{P}(G_{\mathcal{S}} \text{ has density} \geq \rho - \epsilon/2 | H_1) &= \mathbb{P}(G_{\mathcal{S}} \text{ has density} \geq \rho - \epsilon/2 | H_1, G_{\mathcal{S}} = G_{\mathcal{S}}^*)\mathbb{P}(G_{\mathcal{S}} = G_{\mathcal{S}}^*) \\
&\quad + \mathbb{P}(G_{\mathcal{S}} \text{ has density} \geq \rho - \epsilon/2 | H_1, G_{\mathcal{S}} \neq G_{\mathcal{S}}^*)\mathbb{P}(G_{\mathcal{S}} \neq G_{\mathcal{S}}^*) \\
&\geq \mathbb{P}(G_{\mathcal{S}}^* \text{ has density} \geq \rho - \epsilon/2 | H_1)\mathbb{P}(G_{\mathcal{S}} = G_{\mathcal{S}}^*) \\
&\geq (1 - \delta)(1 - \alpha_1),
\end{aligned}
$$

where the second inequality is due to $(E.4)$ and $(E.5)$. This completes the proof of Theorem 3.3.

# F  Additional Simulations

In this subsection, we consider the simulations for the K-colorability test.

Figure 3: Left: A 4-colorable Grotzsch graph. Right: a graph with 3 blocks of Grotzsch graph.

**Settings:** We investigate the type-I and type-II errors of our test via a famous 4-colorable graph called "Grotzsch graph". According to Grotzsch's theorem, every triangle-free planar graph can be colored with only three colors. The Grotzsch graph is constructed to show that the requirement of planarity is necessary in Grotzsch's theorem. The Groetzsch graph is a triangle-free graph with 11 vertices and 20 edges and is 4-colorable but not 3-colorable. The left plot of Figure 3 shows the Groetzsch graph as well as one coloring choice of 4 colors.

Denote the adjacency matrix of the Groetzsch graph as $\mathbf{B} \in \mathbb{R}^{11 \times 11}$ with its $(i,j)$-th entry 1 if node i and node j are connected in the graph, and 0 otherwise. We construct the block diagonal matrix $\mathbf{A}_d \in \mathbb{R}^{d \times d}$ as the block diagonal matrix with each block as $\mathbf{B}$. For example,

$$\mathbf{A}_{33} = \begin{pmatrix} \mathbf{B} & \mathbf{0} & \mathbf{0} \\ \mathbf{0} & \mathbf{B} & \mathbf{0} \\ \mathbf{0} & \mathbf{0} & \mathbf{B} \end{pmatrix},$$

and the graph of $\mathbf{A}_{33}$ is shown in the right plot of Figure 3. For each of 100 repetitions, we use the precision matrix $\mathbf{\Theta}^*(\rho) = \mathbb{1}_d + \rho \mathbf{A}_d$ and generate i.i.d. samples $\mathbf{x}_i \sim N(0, (\mathbf{\Theta}^*(\rho))^{-1})$ for $i = 1, \ldots, n$. We consider various settings with

$$n \in \{400, 800\}, d = 33, \rho \in \{0.25, 0.3, 0.35\}.$$

**Methods:** We compare our algorithm, denoted as **Ours**, with the Direct_node method which estimates the whole precision matrix via nodewise regression first and then test the connectivity on the estimated graphs. As far as we know, no other algorithms are available to handle such problem.

**Result 1 (Computational Costs):** Table 2 summarizes the computational time of our proposed method as well as the Direct_node method. We report both the average time and the maximal time over 100 replicates. Clearly, our method takes much less time than the Direct_node method across all scenarios. In the worse case scenario (maximal time), the Direct_node method takes about 3398 seconds to compute one replicate while our method only takes 5 seconds when $\rho = 0.3$ and $n = 400$.

This indicates the superior computational improvements in this exponential-time example. It is worthy noting that the Direct_node method is not able to output the results within reasonable time when $d = 55$ and hence examples with larger $d$ is not included in this K-colorable testing. In one replicate of $d = 55$, it takes $37854.2$ seconds ($10.5$ hours) for the Direct_node to compute its power. On the other hand, our method takes on average $89$ seconds to compute one replicate.

Table 2: Averaged computational time (in seconds) and maximal computational time (in seconds) among 100 replications of two methods for computing the Type-I error in the K-colorable test.

| n | Criterion | Methods | $\rho = 0.25$ | $\rho = 0.3$ | $\rho = 0.35$ |
|---|---|---|---|---|---|
| 400 | Average time | Ours | 3.05 | 2.93 | 2.69 |
| | | Direct_node | 11.95 | 51.52 | 40.38 |
| | Maximal time | Ours | 4.82 | 5.05 | 4.83 |
| | | Direct_node | 151.02 | 3398.24 | 1855.14 |
| 800 | Average time | Ours | 7.89 | 6.54 | 8.87 |
| | | Direct_node | 20.28 | 29.31 | 114.18 |
| | Maximal time | Ours | 16.07 | 10.82 | 21.01 |
| | | Direct_node | 324.84 | 663.07 | 2731.28 |

**Result 2 (Type-I Error and Power):** Table 3 report the Type-I error as well as the Power analysis of all methods in various scenarios of the K-colorable test.

First, our type-I error is always 0 while that of the Direct_node method is very large (at least 0.7). As we discussed in the connectivity testing example, the Direct_node method is very conservative in variable selection and hence it tends to include many noisy edges. Therefore, it always tends to reject the null K-colorable hypothesis. Hence, the Direct_node method will have a large power. Our power is low when the signal strength is small. However, our power increases to 1 when $\rho$ increases to $0.35$ or sample size increases to 800. Our power loss is expected and is due to both estimation procedure and the sampling procedure as we show in the theorem. The partial power loss due to sampling procedure indicates a tradeoff in the computational cost and the inferential efficiency.

Second, we illustrate the power of our method in the example of $n = 400$ with varying ratio of adding edges. The Groetzsch graph $\mathbf{B}$ is known to be 4-colorable but not 3-colorable. In order to investigate the behavior of our method when distance $\epsilon$ in the alternative hypothesis changes, we randomly add some percentages (denoted as $\eta$) of edges to $\mathbf{B}$ to make it even far away from a 3-colorable graph. Table 4 shows an interesting interaction of the signal level $\rho$ and $\eta$. For a fixed $\eta$, our power increases as $\rho$ increases, and for a fixed $\rho$, our power also increases as $\eta$ increases.

Table 3: Power and Size of two methods in the K-colorable test.

| n | Criterion | Methods | $\rho = 0.25$ | $\rho = 0.3$ | $\rho = 0.35$ |
|---|---|---|---|---|---|
| 400 | Type-I Error | Ours | 0 | 0 | 0 |
| | | Direct_node | 0.93 | 0.85 | 0.73 |
| 400 | Power | Ours | 0 | 0.42 | 1 |
| | | Direct_node | 1 | 1 | 1 |
| 800 | Type-I Error | Ours | 0 | 0 | 0 |
| | | Direct_node | 0.9 | 0.82 | 0.7 |
| 800 | Power | Ours | 0.98 | 1 | 1 |
| | | Direct_node | 1 | 1 | 1 |

Table 4: Power of our method in the K-colorable test with $n = 400$ and varying ratios of adding edges. We randomly add some percentages (denoted as $\eta$) of edges to the Groetzsch graph $\mathbf{B}$ to make it even far away from a 3-colorable graph.

| Criterion | $\rho$ | $\eta = 2\%$ | $\eta = 3\%$ | $\eta = 4\%$ |
|---|---|---|---|---|
| | 0.25 | 0.03 | 0.16 | 0.28 |
| Power | 0.3 | 0.68 | 0.75 | 0.91 |