[Reviews · NeurIPS 2018]

Reviewer 1



In the paper "Sketching Method for Large Scale Combinatorial Inference", the authors introduced a combinatorial inferential framework for graphical models. The new framework provides both the statistical validity and computational efficiency. The results in the paper is for Gaussian graphical model (rather than Markov random fields). The three large-scale combinatorial inference examples are connectivity test, Bipartiteness/K-colorability test, and fast max clique test. Additionally, the paper also discusses the trade-off between the statistical validity and the computation complexity. Overall, the paper is well-written and provides sufficient theoretical results. One question I have is that whether the results in current paper (as in Gaussian graphical model) can be generalized to Markov random fields. I have read the response from the authors. Their response to my question is satisfactory. I would like to keep my original scores.

Reviewer 2



This paper proposes statistical hypothesis testing methods for combinatorial structures of the precision matrix of a Gaussian graphical model. Two types of tests are considered: connectivity test and k-colorability test. In the connectivity test, the proposed method subsamples the nodes and expands the neighborhoods of the sampled nodes. Connectivity is rejected if such an expansion is contained locally. The existence of edge is estimated only when necessary in order to reduce computation cost. In the k-colorability test, the method first estimates the entire precision matrix and apply classical deterministic tests on a thresholded estimated graph. One concern is that the formulation of null and alternative is unnatural as a statistical hypothesis test. In statistical hypothesis testing, the null hypothesis typically refer to a null model with no signal, but the null hypothesis considered in this paper is highly structured: a disconnected precision matrix with bounded degree! The second concern is the stringent conditions required for the validity: the data is Gaussian, and the precision matrix is $s$-sparse. I wonder how $s$ is determined for the real data example. The procedures essentially combines known deterministic tests with standard subsampling with estimated precision matrix, and hence are valid and not surprising. #### Response to author feedback ##### Thanks for your careful explanation. The concern of the usefulness of the hypotheses still remains. In particular, in the connectivity test, the formulation of the null hypothesis seems to be motivated more from the procedure rather than practical interests. In the max clique test, it is unclear how the null and alternative hypotheses reflect "structure" and "structureless".

Reviewer 3



This paper introduces sketching based hypothesis tests for some combinatorial properties of Gaussian graphical models. The authors introduce two sketching procedures --- neighborhood sketching and subgraph sketching, and establish that these procedures control FWER, while being computationally efficient. They supplement their theoretical investigations with simulation experiments and carry out a real data analysis. The Pros: 1. The article attempts to bring together two disparate lines of research--- developments in minimax global testing and multiple testing communities in Statistics on one hand, and Property testing problems in the theoretical computer science literature on the other. I feel this line of research would be of interest to the community, and could lead to interesting future developments. 2. The authors carry out detailed simulation experiments and real data analyses to support their methods. I feel this goes a long way in making their results applicable in real problems. The Cons. 1. The authors argue that testing for combinatorial structures in Gaussian graphical models is a central problem in many applications. While they provide some support for the max-clique problem in Section 5, they do not motivate the other problems studied in this article. I suggest the authors provide some other potential applications for the problems introduced. 2. I was left very confused by some conventions in this article--- principally due to concurrent use of terminology used in multiple testing and property testing literatures. For example, L82 introduces the general schematic for the problems to be studied, but the K-colorable problem (L137) flips the two hypotheses. Coupled with this, in Algorithm 1, the decision to ACCEPT corresponds to rejection of the null hypothesis in traditional statistics literature. This convention makes it very hard to read some sections of the paper. While I understand that the paper is interdisciplinary and that it is hard to explain all conventions due to space constraints, it would be very helpful if the authors could include some pointers for the benefit of the readers. 3. The paper would greatly improve with significant editing--- for example, (i) L-26-- Hypothesis testing literature, L-48 --- Property testing literature, L-114- bounded degree etc. (ii) The Section heading for Section 3 should be Statistical Property" (iii) L79 -- an edge is present if $\Theta_{ij} \neq 0$. In conclusion, I believe that the article presents the synthesis of notions introduced in two very distinct research areas, and introduces an interesting line of research for future enquiry. However, the current submission is sometimes hard to read, and the presentation could be significantly improved. The authors have addressed the questions raised above satisfactorily.